DISCOVERY REPORT

# Critically ill COVID-19 patients with neutralizing autoantibodies against type I interferons have increased risk of herpesvirus disease

Idoia Busnadiego[1‡], Irene A. Abela[1,2‡], Pascal M. Frey[2,3], Daniel A. Hofmaenner[4], Thomas C. Scheier[2], Reto A. Schuepbach[4], Philipp K. Buehler[4], Silvio D. Brugger[2]*, Benjamin G. Hale[1]*

1 Institute of Medical Virology, University of Zurich, Zurich, Switzerland, 2 Department of Infectious Diseases and Hospital Epidemiology, University Hospital Zurich, University of Zurich, Zurich, Switzerland, 3 Department of General Internal Medicine, Inselspital, Bern University Hospital, University of Bern, Bern, Switzerland, 4 Institute of Intensive Care Medicine, University Hospital Zurich, University of Zurich, Zurich, Switzerland

‡ These authors share first authorship on this work.
* silvio.brugger@usz.ch (SDB); hale.ben@virology.uzh.ch (BGH)

The Editors encourage authors to publish research updates to this article type. Please follow the link in the citation below to view any related articles.

## Abstract

Autoantibodies neutralizing the antiviral action of type I interferons (IFNs) have been associated with predisposition to severe Coronavirus Disease 2019 (COVID-19). Here, we screened for such autoantibodies in 103 critically ill COVID-19 patients in a tertiary intensive care unit (ICU) in Switzerland. Eleven patients (10.7%), but no healthy donors, had neutralizing anti-IFNα or anti-IFNα/anti-IFNω IgG in plasma/serum, but anti-IFN IgM or IgA was rare. One patient had non-neutralizing anti-IFNα IgG. Strikingly, all patients with plasma anti-IFNα IgG also had anti-IFNα IgG in tracheobronchial secretions, identifying these autoantibodies at anatomical sites relevant for Severe Acute Respiratory Syndrome Coronavirus 2 (SARS-CoV-2) infection. Longitudinal analyses revealed patient heterogeneity in terms of increasing, decreasing, or stable anti-IFN IgG levels throughout the length of hospitalization. Notably, presence of anti-IFN autoantibodies in this critically ill COVID-19 cohort appeared to predict herpesvirus disease (caused by herpes simplex viruses types 1 and 2 (HSV-1/-2) and/or cytomegalovirus (CMV)), which has been linked to worse clinical outcomes. Indeed, all 7 tested COVID-19 patients with anti-IFN IgG in our cohort (100%) suffered from one or more herpesviruses, and analysis revealed that these patients were more likely to experience CMV than COVID-19 patients without anti-IFN autoantibodies, even when adjusting for age, gender, and systemic steroid treatment (odds ratio (OR) 7.28, 95% confidence interval (CI) 1.14 to 46.31, $p = 0.036$). As the IFN system deficiency caused by neutralizing anti-IFN autoantibodies likely directly and indirectly exacerbates the likelihood of latent herpesvirus reactivations in critically ill patients, early diagnosis of anti-IFN IgG could be rapidly used to inform risk-group stratification and treatment options.

**Trial Registration:** ClinicalTrials.gov Identifier: NCT04410263.

**Data Availability Statement:** All relevant data are within the paper and its Supporting Information files.

**Funding:** Work in the BGH laboratory is funded by the Swiss National Science Foundation through grant 31003A_182464 to BGH. IAA is supported by a research grant from the Promedica Foundation (#14851M/1). SDB has received funding from the USZ Foundation through grant USZF270808. The funders had no role in study design, data collection and analysis, decision to publish, or preparation of the manuscript.

**Competing interests:** The authors have declared that no competing interests exist.

**Abbreviations:** APS-1, autoimmune polyendocrine syndrome type I; CI, confidence interval; CMV, cytomegalovirus; COVID-19, Coronavirus Disease 2019; FCS, fetal calf serum; HSV-1, herpes simplex virus type 1; ICU, intensive care unit; IFN, interferon; MFI, median fluorescence intensity; OR, odds ratio; PE, phycoerythrin; RT-PCR, reverse transcription PCR; SARS-CoV-2, Severe Acute Respiratory Syndrome Coronavirus 2; SD, standard deviation; TBS, tracheobronchial secretion; VZV, varicella-zoster virus.

## Introduction

Deficiencies in the human antiviral type I interferon (IFN) system can predispose individuals to severe viral disease, most notably during infections with antigenically novel pathogens to which preexisting humoral immunity is lacking [1]. The ongoing Severe Acute Respiratory Syndrome Coronavirus 2 (SARS-CoV-2) pandemic has highlighted a previously unappreciated type of functional IFN deficiency mediated by autoantibodies that neutralize the action of several type I IFNs, particularly the IFNα or IFNω subtypes [2], and rarely IFNβ [3]. Across multiple independent studies, around 10% of critically ill Coronavirus Disease 2019 (COVID-19) patients, but not those with very mild infections, have serum autoantibodies that inhibit the antiviral function of IFNα and/or IFNω in vitro [2–11]. Furthermore, presence of anti-IFN autoantibodies has been associated with 20% of all COVID-19 deaths, and this has disproportionately affected older individuals [3,12]. For example, serum autoantibodies targeting IFNα and/or IFNω have been found in a very low proportion (0.17%) of healthy individuals under 70 years of age, but their prevalence is increased in the elderly such that prevalence is around 4% in those over 70 [3,7]. The presence of these autoantibodies in prepandemic samples taken from some individuals who later developed severe COVID-19 suggests that SARS-CoV-2 infection is not directly responsible for their production, but that their presence might predispose to more critical illness [2,3,6,7].

Importantly, anti-IFN autoantibodies have also been detected in nasal swabs and bronchoalveolar lavages of severe COVID-19 patients [13,14]. The presence of neutralizing autoantibodies targeting type I IFNs is thereby associated with lower levels of IFN-dependent antiviral gene expression signatures in nasal mucosa as well as immune cell dysfunction [2,5–7,13,15]. These functional consequences likely permit higher and persistent SARS-CoV-2 viral loads in patient nasopharynges, which may potentiate the excessive inflammation that drives some forms of critical disease with this respiratory infection [2,5–7,13,15]. Koning and colleagues also demonstrated that critically ill COVID-19 patients with neutralizing anti-IFN autoantibodies more frequently display additional severe clinical complications, such as renal failure, bacterial pneumonia, and thromboembolic events [5]. Thus, exacerbated SARS-CoV-2 replication in respiratory tissues alone may not fully explain the contributions of anti-IFN autoantibodies to severe COVID-19 and other systemic pathogenic mechanisms may occur. Notably, concomitant herpesvirus (e.g., herpes simplex virus type 1 (HSV-1), cytomegalovirus (CMV), varicella-zoster virus (VZV)) reactivations have been recognized to be associated with more severe disease and worse clinical outcomes in critically ill COVID-19 patients [16]. However, despite the importance of a functional IFN system in maintaining herpesvirus latency in experimental settings [17–19], potential associations between the presence of anti-IFN autoantibodies, herpesvirus reactivations, and clinical outcomes in critically ill patients have yet to be investigated.

In this study, we sought to evaluate the prevalence of autoantibodies (IgG, IgM, and IgA) targeting and neutralizing type I IFNs in a longitudinally sampled cohort of 103 critically ill COVID-19 patients as compared to healthy controls. Furthermore, we aimed to describe variation in COVID-19 disease severity in patients with anti-IFN autoantibodies and perform exploratory analyses to investigate whether the presence of anti-IFN autoantibodies correlated with herpesvirus reactivation and disease.

## Methods

### Cohort description

The study was conducted as part of the MicrobiotaCOVID cohort study [20], a single-center, prospective observational study conducted at the Institute of Intensive Care Medicine of the

University Hospital Zurich, Switzerland together with the Department of Infectious Diseases and Hospital Epidemiology, University Hospital Zurich, Switzerland and registered at clinicaltrials.gov (ClinicalTrials.gov Identifier: NCT04410263). We enrolled 103 patients with COVID-19 ARDS (CARDS) who were admitted to the intensive care unit (ICU) between March 2020 and April 2021. The study was approved by the Local Ethics Committee of the Canton of Zurich, Switzerland (Kantonale Ethikkommission Zurich BASEC ID 2020–00646) in accordance with the provisions of the Declaration of Helsinki and the Good Clinical Practice guidelines of the International Conference on Harmonisation. All data were analyzed anonymously.

### Healthy controls

Plasma samples from 130 anonymized prepandemic healthy adults were derived from specimens provided by the Zurich Blood Transfusion Service of the Swiss Red Cross for a previous study [21] and were used with approval of the responsible Local Ethics Committee of the Canton of Zurich, Switzerland (Kantonale Ethikkommission Zurich BASEC ID 2021–00437 and 2021–01138).

### Data collection and covariates

Clinical and laboratory data were obtained as previously described [20].

### Sample collection, processing, and testing (virus diagnostics)

SARS-CoV-2 was detected by real-time reverse transcription PCR (RT-PCR) as previously described [20]. Moreover, we assessed serum detection and viral load of the following herpesviruses, also as previously described [20]: herpes simplex viruses type 1 and 2 (HSV-1 and -2), CMV, and VZV. Virus diagnostics were initiated by the treating physicians according to the clinical situation and were not performed systematically in all patients. Herpesvirus disease was defined as detection of HSV-1/2, CMV, or VZV in blood by PCR and/or a clinical manifestation with PCR confirmation in a corresponding sample (i.e., herpes labialis, herpes zoster, tracheobronchitis, mucositis including stomatitis, and genital manifestations).

### Sample processing and testing (IFN-binding antibodies)

A high-throughput bead-based serological assay was established using methods adapted from a previous study [22] (**S1A Fig**). Briefly, magnetic beads (MagPlex-C Microspheres, Luminex) were coupled to recombinant human IFNs (IFNα2: Novusbio NBP2-35893, IFNβ: Peprotech 300-02BC, or IFNω: Novusbio NBP2-34971) or albumin (Sigma-Aldrich 70024-90-7) at a concentration of 10 µg protein per million beads. Bead coating was assessed using mouse monoclonal antibodies against IFNα2, IFNβ, or IFNω (anti-IFNα2: Novusbio NB100-2479, anti-IFNβ: pbl assay science 21465–1, anti-IFNω: Novusbio NBP3-06154). Patient samples were diluted 1:50 in PBS supplemented with 1% BSA (PBS/BSA) and incubated with 1:1:1:1 mixtures of the coated beads for 1 h at room temperature. As a positive control, a human polyclonal anti-IFNα2b antiserum was used (BEI resources: NR-3072). Beads were washed twice with PBS/BSA before phycoerythrin (PE)-labeled secondary antibodies were added separately at a 1:500 dilution in PBS/BSA (Southern Biotech: IgA 205009, IgM 202009, IgG 204009, mouse IgG: BioLegend 405307). After 1 h incubation at room temperature, bead mixtures were washed twice in PBS/BSA, and samples were analyzed on a FlexMap 3D instrument (Luminex). A minimum of 50 beads per antigen were acquired. Median fluorescence intensity (MFI) values from the IFN-coated beads were obtained and calculated relative to the MFI

obtained from albumin-coated beads. For each isotype, the mean and standard deviations (SDs) were calculated from the MFI values obtained from the healthy donor samples, and MFI values above 10 SDs for IFNα2, or 5 SDs for IFNω were considered positive. The higher stringency for IFNα2 was chosen because of its lower background variability with healthy donor samples.

In validation experiments, the mouse monoclonal IgG antibody targeting human IFNα2 (clone ST29) exhibited specific reactivity to beads coated with human IFNα2 (as compared to beads coated with albumin), but showed some cross-reactivity to beads coated with human IFNω, and no cross-reactivity to beads coated with human IFNβ (**S1B Fig**). Similarly, the mouse monoclonal IgG antibody targeting human IFNβ (clone MMHB-15) was highly specific to beads coated with human IFNβ, and gave no reactivity to beads coated with IFNα2, but showed some cross-reactivity to beads coated with human IFNω (**S1C Fig**). The mouse monoclonal IgG antibody targeting human IFNω (clone 04) was highly specific to beads coated with human IFNω, and gave no reactivity to beads coated with either IFNα2 or IFNβ (**S1D Fig**). When the assay was validated with human samples, pooled sera from a panel of 20 healthy donors exhibited no IgG binding to either IFNα2 or IFNβ beads, but some low reactivity to IFNω beads (**S1E Fig**, left panel). The human polyclonal anti-IFNα2b antiserum had IgG that strongly reacted with the IFNα2 beads, and to some extent the IFNω beads, but not with the IFNβ beads (**S1E Fig**, right panel). Cross-reactivity of IgG antibodies against IFNα2 and IFNω is expected given the close relatedness of these type I IFNs and previous descriptions of cross-reactive monoclonal antibodies [2,23].

## Sample processing and testing (IFN-neutralizing antibodies)

Approximately $2.4 \times 10^4$ human embryonic kidney HEK293T cells (ATCC CRL-3216) per well in 96-well plates were reverse-transfected with 30 ng of a plasmid containing the firefly luciferase (FF-Luc) gene under control of the IFN-inducible mouse *Mx1* promoter (pGL3-Mx1P-FFluc) (kindly provided by Georg Kochs), together with 4 ng of a control plasmid expressing *Renilla* luciferase (Ren-Luc) under a constitutively active promoter (pRL-TK-Renilla). Cells were transfected using FuGene HD (Promega E2311) and incubated at 37°C and 5% $CO_2$ in Dulbecco's Modified Eagle medium (DMEM, #41966–029, Gibco) supplemented with 10% (v/v) fetal calf serum (FCS), 100 U/mL penicillin, and 100 mg/mL streptomycin (#15140–122: Gibco). Twenty-four hours post-transfection, patient plasma samples were diluted 1:50 in DMEM supplemented with 10% FCS, 100 U/mL penicillin, and 100 mg/mL streptomycin and incubated for 1 h at room temperature with 10, 1, or 0.2 ng/mL of IFNα2 or IFNω prior to their addition to transfected cells. After 24 h, cells were lysed for 15 min at room temperature, and FF-Luc and Ren-Luc activity levels were determined using the Dual-Luciferase Reporter Assay System (E1960, Promega) and a PerkinElmer EnVision plate reader (EV2104) according to the manufacturers' instructions. FF-Luc values were normalized to Ren-Luc values and then to the median luminescence intensity of control wells that had not been stimulated with either IFNα2 or IFNω.

## Statistical analyses

The association of anti-IFN autoantibodies and herpesvirus disease, with the additional analyses of only CMV or HSV-1/2, was examined using logistic regression models. All 3 models were adjusted for age, gender, and treatment with systemic corticosteroids. Only COVID-19 ICU patients tested for herpesviruses by PCR (59 out of 103) were included in these analyses. Analyses were performed using SPSS Version 23 (SPSS Science, Chicago, Illinois, United States of America) and Stata 16 (Stata Corporation, College Station, Texas, USA).

## Results

### Autoantibodies targeting type I IFNs in the plasmas of critically ill COVID-19 patients

We used a multiplexed bead-based assay to screen for autoantibodies targeting representative type I IFNs (IFNα2, IFNβ, and IFNω; [2]) in a cohort of 103 individuals (179 samples from 80 males, aged 31 to 81; and 51 samples from 23 females, aged 20 to 87; overall median age 66) who were admitted to the ICU of the University Hospital Zurich with severe COVID-19 between March 2020 and April 2021 (Table 1). Plasma samples from 130 prepandemic healthy adults (75 males, aged 19 to 70 and 55 females, aged 19 to 69) were used as a negative control group to set benchmark thresholds. We observed that 11.3% of male severe COVID-19 patients (9/80 individuals, 15 samples) and 13.0% of female severe COVID-19 patients (3/23 individuals, 8 samples) had clearly detectable IgG autoantibodies targeting IFNα2 in their plasma, which were not present in the plasma of 130 healthy donors (Fig 1A). Anti-IFNα2 autoantibodies were largely confined to the IgG class, as few individuals had IgA or IgM reactivity to IFNα2, although 1 male COVID-19 patient (2 samples) was highly positive for anti-IFNα2 IgM (Fig 1A). We similarly detected prevalent reactivity of autoantibodies against IFNω, with detectable IgG autoantibodies in 7.5% of male severe COVID-19 patients (6/80 individuals, 8 samples) and 8.7% of female severe COVID-19 patients (2/23 individuals, 6 samples), although these estimates may be on the low side due to some heterogeneity in the healthy donors (Fig 1B), which is consistent with our observed background reactivity to the IFNω-coated beads (S1E Fig). Notably, all anti-IFNω IgG positive samples were also anti-IFNα2 IgG positive, but 10 of the anti-IFNα2 IgG positive samples (7 patients) were negative for anti-IFNω IgG. While IgA autoantibodies targeting IFNω were observed in a few individuals, it was striking that 5% of males (4/80 individuals, multiple samples) were positive for anti-IFNω IgM, which could be suggestive of recent induction of these anti-IFNω antibodies (Fig 1B).

**Table 1. Patient baseline characteristics.**

| Characteristic[#] | All patients (*n* = 103) | Patients with anti-IFN autoAbs (*n* = 12) | Patients without anti-IFN autoAbs (*n* = 91) |
|---|---|---|---|
| | *n* (%) or median (interquartile range) | | |
| Age, years | 66 (56–71) | 68 (59–74) | 66 (55–70) |
| Female gender | 23 (22) | 3 (25) | 20 (22) |
| Body mass index, kg/m$^2$ | 28 (24–32) | 30 (25–34) | 28 (24–31) |
| Systemic corticosteroids | 87 (85) | 11 (92) | 76 (84) |
| SAPS II | 37 (29–50) | 39 (31–52) | 37 (29–49) |
| SOFA Score | 7 (3–9) | 7 (3–8) | 7 (4–10) |
| Arterial hypertension | 50 (49) | 6 (50) | 44 (48) |
| Cardiac disease | 40 (39) | 6 (50) | 34 (37) |
| Cerebrovascular disease | 16 (16) | 1 (8) | 15 (16) |
| Chronic liver disease | 6 (6) | 0 (0) | 6 (7) |
| Chronic renal disease | 17 (17) | 0 (0) | 17 (19) |
| COPD | 9 (9) | 1 (8) | 8 (9) |
| Diabetes mellitus | 31 (30) | 4 (33) | 27 (30) |
| History of cancer | 12 (12) | 1 (8) | 11 (12) |
| Immunosuppression | 34 (33) | 4 (33) | 30 (33) |
| Solid organ transplant | 9 (9) | 1 (8) | 8 (9) |

[#]Data underlying this table can be found in S1 Data.

autoAbs, autoantibodies; COPD, chronic obstructive pulmonary disease; IFN, interferon; SAPS, Simplified Acute Physiology Score; SOFA, Sequential Organ Failure Assessment.

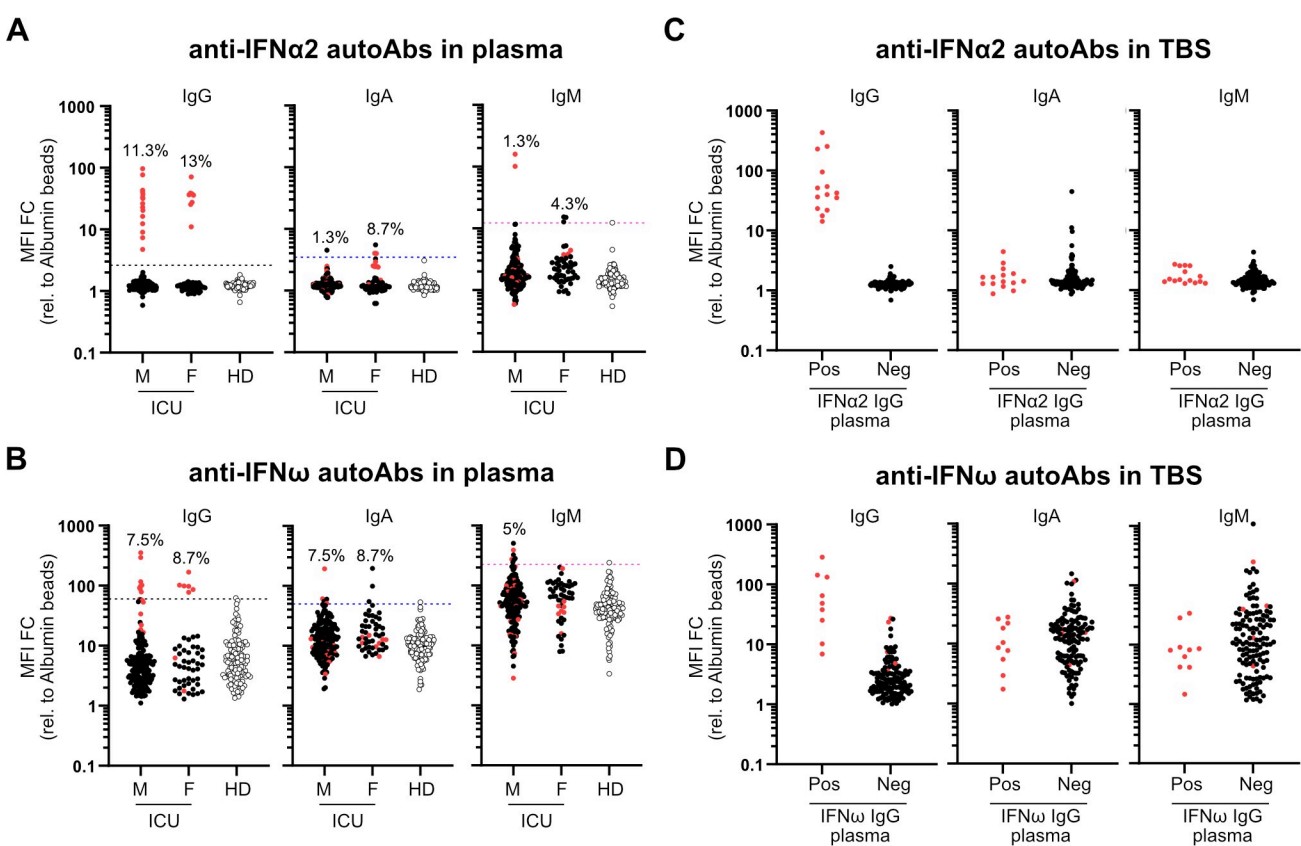

**Fig 1. Autoantibodies targeting type I IFNs in the plasmas and tracheobronchial secretions of critically ill COVID-19 patients. (A and B)** Multiplexed bead-based assay to detect IgG, IgA, and IgM autoAbs against IFNα2 (A) or IFNω (B) in plasmas of patients in ICU with severe COVID-19 (Male: M = 179 samples corresponding to 80 patients, Female: F = 51 samples corresponding to 23 patients) or Healthy Donors (HD = 130 samples). MFI FC of signal derived from IFN-coated beads relative to the MFI of signal derived from albumin-coated beads is shown. Dashed lines indicate 10 SDs (A) or 5 SDs (B) from the mean calculated from HD values for each IFN and each isotype. Values above the dashed lines are considered positive. Percentage of positive patients (not samples) per analyzed group is indicated. (C and D) Multiplexed bead-based assay to detect IgG, IgA, and IgM autoAbs against IFNα2 (C) or IFNω (D) in TBSs of COVID-19 ICU patients described in (A). Pos (positivity) and Neg (negativity) for anti-IFNα2 IgG (C) or anti-IFNω IgG (D) in plasma samples from the same patient were used to stratify patients. MFI FC of signal derived from IFN-coated beads relative to the MFI of signal derived from albumin-coated beads is shown. In all panels, red dots indicate the patients/samples that were positive for anti-IFNα2 IgG autoAbs in plasma (A) and are denoted simply for reference. Data underlying this figure can be found in S1 Data. autoAbs, autoantibodies; COVID-19, Coronavirus Disease 2019; FC, fold change; HD, Healthy Donors; ICU, intensive care unit; IFN, interferon; MFI, median fluorescence intensity; SD, standard deviation; TBS, tracheobronchial secretion.

We did not identify any patients who were unambiguously positive for anti-IFNβ autoantibodies. The identification of anti-IFN autoantibodies in approximately 10% of severe COVID-19 patients is fully in line with previous reports from others [2–11].

### Autoantibodies targeting type I IFNs in tracheobronchial secretions of critically ill COVID-19 patients

In order to have direct functional consequences for SARS-CoV-2 replication, autoantibodies targeting type I IFNs would have to be present in either the nasopharynges or tracheal tracts, as recently demonstrated [13,14]. We therefore used our antibody-binding assay to assess anti-IFNα2 and anti-IFNω IgG, IgA, and IgM autoantibodies in tracheobronchial secretions (TBSs) obtained from 88 of the severe COVID-19 patients in our cohort. Stratifying by plasma IgG positivity to either IFNα2 or IFNω, it was clear that patients with detectable plasma IgG to type I IFNs (particularly IFNα2) also had detectable anti-IFN IgG autoantibodies in their TBSs

(**Fig 1C** and **1D**). In contrast, we did not readily detect either IgA or IgM anti-IFN autoanti-bodies in TBSs. Assessment of anti-IFNβ autoantibody levels did not reveal differences between severe COVID-19 patients who were positive or negative for anti-IFNα2 IgG, either in plasma or TBS samples (**S2A** and **S2B Fig**). These data indicate that at least anti-IFNα2 and anti-IFNω IgG autoantibodies are present at the physiological sites of SARS-CoV-2 replication (i.e., the trachea) where they are most likely to exert functional relevance during infection.

## Longitudinal analysis of autoantibodies targeting type I IFNs in individual critically ill COVID-19 patients

For 9 of the 12 severe COVID-19 patients who were positive for anti-IFNα2 IgG autoantibod-ies, we had multiple plasma samples that were collected around the time of ICU admission, ICU discharge, or at hospital discharge. These samples spanned between 8 and 58 days post-admittance to ICU and revealed different patterns of anti-IFN autoantibody levels. For exam-ple, in some individuals, the levels of anti-IFN autoantibodies appeared to reduce over time, while in others, levels increased or remained constant (**Fig 2A** and **2B**). A recent report has also noted either stable or fluctuating levels of anti-IFNα2 IgG autoantibodies following hospi-tal admission for COVID-19 [7]. Given our lack of "baseline" samples from individuals prior to their infection with SARS-CoV-2, and the fact that our earliest samples are from admittance to ICU (likely a late event in disease progression), it is impossible to conclude whether these anti-IFN autoantibodies preexisted in these individuals prior to COVID-19. However, the gen-eral lack of anti-IFN IgM autoantibodies in most individuals, even at these relatively late times, may be suggestive that the autoantibodies preexisted.

## Autoantibodies targeting type I IFNs are mostly neutralizing

To functionally characterize the anti-IFN autoantibodies detectable in patient plasma samples, we adapted a standard cell-based luciferase reporter assay that relies on IFN-stimulated activation of the IFN-inducible *Mx1* promoter (**Fig 2C**). Notably, 21/23 patient plasmas with detectable anti-IFNα2 IgG autoantibodies were able to neutralize the function of IFNα2 in this assay, irrespective of whether a low concentration of IFNα2 (0.2 ng/mL) or a high concentration of IFNα2 (10 ng/mL) was used (**Fig 2D**). Strikingly, 2/23 patient plasmas (both originating from patient 37, a female in her 70s) did not exhibit neutralization capabilities at any of the IFNα2 concentrations tested, despite having higher IFNα2-binding IgG titers than many other samples that did neutral-ize IFNα2 (**Fig 2D**). Similar data were obtained when neutralization of IFNω was assessed, although differences were noted (**Fig 2D**). For example, some samples from patient 19 (a male in his 60s) and patient 31 (a male in his 50s) did not have detectable anti-IFNω-binding IgG autoan-tibodies, though they could efficiently neutralize low IFNω concentrations (0.2 and 1 ng/mL), but not high IFNω concentrations (10 ng/mL), possibly due to cross-reactive anti-IFNα2-binding antibodies present in the samples or differences in sensitivity between the binding and neutraliza-tion assays. In this regard, although we only assessed neutralization for samples with detectable IFN binding autoantibodies, a recent report indicates that assaying IFN neutralization can increase the detection of functionally relevant anti-IFN autoantibodies because such assays are likely to be more sensitive to much lower concentrations of IFN [3].

## Presence of autoantibodies targeting type I IFNs as a predictor of herpesvirus disease

We assessed links between having anti-IFN autoantibodies and several patient characteristics associated with worse clinical outcomes in COVID-19 patients admitted to ICU. In the context

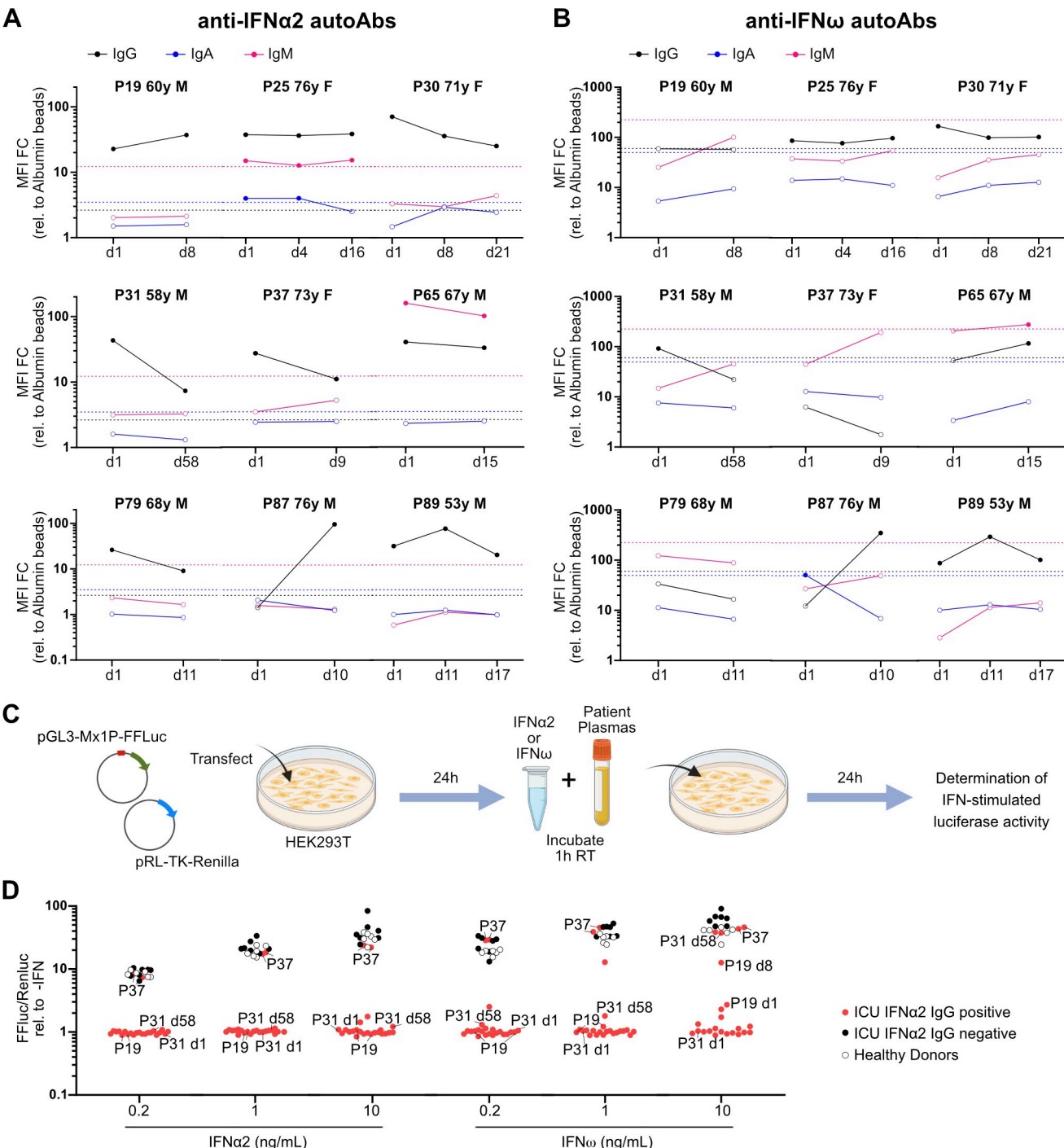

**Fig 2. Longitudinal analysis of plasma autoantibodies targeting type I IFNs in individual critically ill COVID-19 patients and their neutralization capacities.** (**A** and **B**) Longitudinal analysis of plasma anti-IFNα2 (A) and anti-IFNω (B) IgG, IgA, and IgM autoAbs in selected critically ill COVID-19 patients positive for plasma anti-IFNα2 IgG. Samples were collected on day of admission to ICU (d1) and as indicated thereafter. MFI FC of signal derived from IFN-coated beads relative to the MFI of signal derived from albumin-coated beads is shown. Dashed lines indicate 10 SDs (IFNα2) or 5 SDs (IFNω) from the mean calculated from HD values for each IFN and each isotype in Fig 1A and are used as threshold values for positivity (filled circles). Internal patient identifier numbers (P) are shown, together with the individual's gender (male, M; female, F) and age (years, y). (**C**) Schematic representation of the luciferase reporter-based neutralization assay. HEK293T cells are cotransfected with a pGL3-Mx1P-FFLuc reporter (FF-Luc) plasmid and a constitutively active pRL-TK-Renilla (Ren-Luc) plasmid. After 24 h, cells are incubated with IFNα2 or IFNω that have been preincubated with patient plasmas. After a further 24 h, cells are lysed, and IFN-stimulated luminescence intensity (FF-Luc) is measured and made relative to the constitutively active Ren-Luc. Schematic created with BioRender.com. (**D**) Results for the neutralization of 10, 1, or 0.2 ng/mL of IFNα2 or IFNω in the presence of 1/50 diluted patient

plasmas from ICU COVID-19 patients positive for anti-IFNα2 IgG ($n = 12$), ICU COVID-19 patients negative for anti-IFNα2 IgG ($n = 6$), or HD ($n = 6$). FF-Luc values were made relative to Ren-Luc values and then normalized to the median luminescence intensity of control samples without IFN. Some individual patient (P) and sampling day (d) identifiers (corresponding to Fig 2A and 2B) are shown for comparison with their IFNα2 or IFNω binding data. Data underlying this figure can be found in S1 Data. autoAbs, autoantibodies; COVID-19, Coronavirus Disease 2019; FC, fold change; HD, Healthy Donors; IFN, interferon; MFI, median fluorescence intensity; SD, standard deviation.

of patient baseline characteristics, we could not observe any attributes that clearly correlated with the presence of anti-IFNα IgG autoantibodies, including age (median age of patients with anti-IFN autoantibodies was 68 as compared to 66 for patients without anti-IFN autoantibodies), gender, body mass index, or several chronic underlying conditions, such as diabetes, cancer, or cardiac, liver, and renal diseases (Table 1). Furthermore, we were unable to observe any clear association between presence of anti-IFN autoantibodies and outcomes such as death, length of hospitalization, length of ICU stay, or duration of ventilation (Table 2). We therefore assessed more quantifiable parameters that can impact disease outcomes in ICU, such as the prevalence of bacterial superinfections in the blood or respiratory tract, and the levels of herpesviruses such as HSV-1/2, CMV, and VZV in the blood. While we were unable to find an association between presence of anti-IFN autoantibodies and the likelihood of bacterial superinfections (Table 2), it was notable that presence of anti-IFN autoantibodies was a clear predictor of herpesvirus disease (Table 3 and Fig 3). Specifically, in our cohort of 103 patients, a subset of 59 individuals (57%) were tested for HSV-1/2, VZV, or CMV by blood PCR according to the treating physician's decision and clinical manifestations such as herpes labialis, herpes zoster, tracheobronchitis, mucositis (including stomatitis), or anogenital lesions. In most cases, PCR testing was performed independently of prior knowledge on herpesvirus serostatus. Of these 59 patients, herpesvirus infections were confirmed by PCR in 38 (64%) patients, consisting of 30 (51%) patients with HSV-1/2, 21 (36%) with CMV, and none with VZV (Table 3). Thirteen (22%) of the 59 patients had both HSV-1/2 and CMV infections. Strikingly, all patients in this subset of patients with anti-IFN autoantibodies ($n = 7$, 100%) experienced herpesvirus disease, although not all patients with herpesviruses also had anti-IFN autoantibodies (Table 3 and Fig 3). For all patients with confirmed herpesvirus infections, viral loads and reported clinical manifestations did not differ notably between individuals with or without anti-IFN autoantibodies. Nevertheless, after adjusting for age, gender, and systemic corticosteroid treatment, patients with anti-IFN autoantibodies were more likely to experience CMV

**Table 2. Description of patient outcomes.**

| Outcome[#] | All patients ($n = 103$) | Patients with anti-IFN autoAbs ($n = 12$) | Patients without anti-IFN autoAbs ($n = 91$) |
|---|---|---|---|
| | | n (%) or median (interquartile range) | |
| Death | 23 (22) | 3 (25) | 20 (22) |
| Death at 28 days | 20 (19) | 3 (25) | 17 (19) |
| Death on ICU | 22 (21) | 3 (25) | 19 (21) |
| Length of hospital stay, days | 27 (16–51) | 23 (17–29) | 27 (16–53) |
| Length of ICU stay, days | 15 (8–28) | 15 (7–27) | 16 (8–29) |
| Duration of ventilation, days* | 12 (7–19) | 17 (9–20) | 11 (7–19) |
| Bacterial superinfection[†] | 35 (34) | 3 (25) | 32 (35) |

*Data only available from 90 patients.

[†]Defined previously [20].

[#]Data underlying this table can be found in S1 Data.

autoAbs, autoantibodies; ICU, intensive care unit; IFN, interferon.

**Table 3. Description of herpesvirus detection.**

| Herpesvirus* | All patients† (n = 59) | Patients with anti-IFN autoAbs (n = 7) | Patients without anti-IFN autoAbs (n = 52) |
|---|---|---|---|
| | | n (%) | |
| Any herpesvirus | 38 (64) | 7 (100) | 31 (60) |
| CMV | 21 (36) | 5 (71) | 16 (31) |
| HSV-1/2 | 30 (51) | 6 (86) | 24 (46) |
| VZV | 0 | - | - |

*Testing done by PCR in blood.

†Only a subset of patients (59 out of 103) were tested for herpesviruses according to local clinical decisions.

autoAbs, autoantibodies; CMV, cytomegalovirus; HSV-1/2, herpes simplex virus types 1 or 2; IFN, interferon; VZV, varicella-zoster virus.

(odds ratio (OR) 7.28, 95% confidence interval (CI) 1.14 to 46.31, $p = 0.036$) or both HSV-1/2 and CMV (OR 8.47, CI 1.37 to 52.31, $p = 0.021$), while results for HSV-1/2 alone were less clear, but certainly suggestive (OR 8.04, CI 0.78 to 82.81, $p = 0.08$). These data indicate that presence of anti-IFN autoantibodies might contribute to herpesvirus disease. Based on the high seroprevalence of HSV and CMV in comparable cohorts in Switzerland, it is likely that reactivation of latent herpesviruses accounts for these clinical conditions.

## Discussion

In this study, we report the presence of IgG autoantibodies that bind and neutralize the type I IFNs, IFNα2 and IFNω, in plasmas/sera and TBSs from approximately 10% of critically ill COVID-19 patients admitted to a tertiary ICU in Switzerland. It is likely that autoantibody binding and neutralization of IFNα2 is a convenient marker for binding and neutralization of most other IFNα subtypes [2]. Our study demonstrates the importance of longitudinal analysis of autoantibodies directed against type I IFNs, as we observed different patterns of anti-IFN autoantibody levels in individual COVID-19 patients over time, although the significance of this is currently unclear. We further establish a potential link between the presence of anti-IFN autoantibodies and possible reactivation of latent virus infections, particularly herpesviruses. Anti-IFN autoantibodies were not detected in any of the healthy donors tested, suggesting an enrichment in critically ill COVID-19 patients that may contribute to the development of severe disease in some individuals. However, we note that (at least within our rather small COVID-19 ICU cohort) presence of anti-IFN autoantibodies was not significantly associated with parameters such as death, length of hospitalization, length of ICU stay, or duration of ventilation. While this broadly contrasts with the findings of others who noted an association between presence of anti-IFN autoantibodies and increased COVID-19 disease severity parameters [2,3,5,7], this difference could be explained by a lack of power in our exploratory study or masking effects of the high standard of care in a high-resource setting. Nevertheless, the proportion of critically ill COVID-19 patients in our cohort with anti-IFN autoantibodies is remarkably consistent with the findings from several independent severe COVID-19 cohorts recently studied across Europe, Asia, and the Americas, despite the use of different detection assays [2–11,24,25]. Indeed, in the future, it will probably be important to have standardized quantitative assays and reporting standards for such anti-IFN autoantibodies, as varying assay sensitivities may mean that their presence is under or overestimated. In particular, it was shown that assaying IFN neutralization, rather than simply binding, increases the detection of functionally relevant anti-IFN autoantibodies because such assays are likely to be sensitive to much lower, potentially more physiologically relevant, concentrations of IFNs [3].

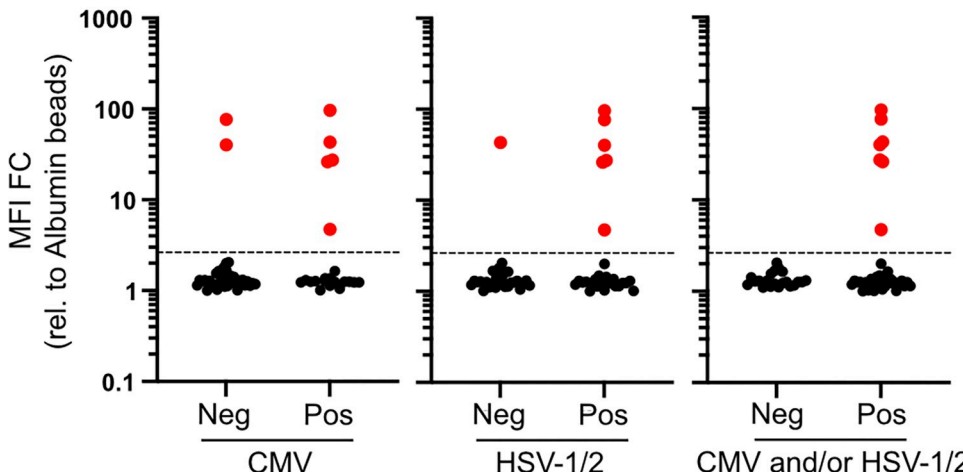

**Fig 3. Presence of autoantibodies targeting type I IFNs as a predictor of herpesvirus disease in critically ill COVID-19 patients.** Fifty-nine severe COVID-19 patients in ICU were tested for herpesvirus (HSV-1/2 and CMV) reactivations in their blood by PCR. Pos (positivity) and Neg (negativity) for CMV, HSV-1/2, and CMV and/or HSV-1/2 were used to stratify the results obtained when plasma samples from the same patient were assayed for IgG autoAbs targeting IFNα2 (see Fig 1). MFI FC of signal derived from IFNα2-coated beads relative to the MFI of signal derived from albumin-coated beads is shown for each individual patient. Values above the dashed line are considered positive (red). Data underlying this figure can be found in S1 Data. autoAbs, autoantibodies; CMV, cytomegalovirus; COVID-19, Coronavirus Disease 2019; FC, fold change; MFI, median fluorescence intensity; HSV-1/2, herpes simplex virus types 1 or 2; ICU, intensive care unit; IFN, interferon.

The most notable clinical feature that we found to be associated with the presence of anti-IFN autoantibodies was the increased probability of herpesvirus disease, which is highly reminiscent of one of the earliest descriptions of anti-IFN autoantibodies identified in a single patient experiencing disease caused by the herpesvirus, VZV [26]. Indeed, all patients tested in our cohort with anti-IFN autoantibodies demonstrated active herpesvirus infections (CMV, HSV-1/2, or both), and thus detection of anti-IFN autoantibodies appears to be an excellent predictor of likely reactivations in our exploratory analysis. Both CMV and HSV-1 reactivations are commonly reported events in patients who have been admitted to ICU, even in those who are otherwise immunocompetent or who have been admitted for noninfectious clinical reasons [27,28]. Furthermore, it is well described that herpesvirus reactivations are associated with worse outcomes in non-COVID patients, with increased length of stay in ICU, increased length of mechanical ventilation, and increased mortality [29–31]. Similarly, HSV-1 and CMV reactivations have been observed in critically ill COVID-19 ICU patients, and herpesvirus reactivations in these patients have been associated with an increased risk of pneumonia and mortality [16,32]. Thus, it could be that anti-IFN autoantibodies are a predisposing factor for pathogenic herpesvirus reactivations in a subset of COVID-19 patients, and this may have important implications for our understanding of the immunologic phenomena underlying severe COVID-19, risk stratification, and of course possible herpesvirus-directed therapeutic options. Future studies would have to investigate whether screening for anti-IFN autoantibodies, and prophylaxis against herpesviruses, can improve clinical outcomes.

Mechanistically, it is currently unclear if the general IFN system deficiency caused by presence of anti-IFN autoantibodies is sufficient to trigger herpesvirus reactivations directly (and thus contribute to disease severity in affected COVID-19 patients) or whether herpesvirus reactivations are an epiphenomenon of severe inflammatory disease caused by uncontrolled

SARS-CoV-2 replication in these patients, perhaps who are then also more likely to be treated with steroids that can increase herpesvirus reactivations [33]. Interestingly, however, our analysis is adjusted for steroid use, suggesting that the substantially increased likelihood of herpesvirus reactivations in those with anti-IFN autoantibodies is independent of systemic steroid treatments. In addition, some evidence may already suggest a direct contributing role of anti-IFN autoantibodies in being causative in triggering herpesvirus reactivations. For example, in a murine model system, just the absence of functional type I IFNs could cause CMV reactivation from latently infected endothelial cells [17]. Similarly, experimental depletion of type I IFNs using neutralizing antibodies led to an increased propensity of murine gammaherpesvirus (MHV-68) reactivation in mice [19]. Adverse herpesvirus reactivations in humans have also been reported following treatment regimens involving tofacitinib or baricitinib (two JAK inhibitors that limit functionality of the IFN system) [34,35]. Furthermore, and most importantly perhaps, a recent study of individuals suffering from autoimmune polyendocrine syndrome type I (APS-1; a genetic disease caused by defects in the *AIRE* gene leading to production of autoantibodies targeting type I IFNs) showed that high levels of neutralizing anti-IFN autoantibodies are associated with herpesvirus (VZV) reactivation and severe clinical outcomes [36]. Individual patients with neutralizing anti-IFNα antibodies and generalized VZV or VZV central nervous system vasculopathy have also been reported [26,37]. Thus, it is highly plausible that the neutralizing anti-IFN autoantibodies that we detect in approximately 10% of critically ill COVID-19 ICU patients can directly contribute to latent herpesvirus reactivations and subsequent disease.

A clear limitation of our study is the low patient sample size in our cohort and single-center study design that did not provide us with sufficient statistical power to allow detection of small differences in clinical outcomes. This could be improved in future studies with higher participant numbers and in studies with a predefined systematic sampling procedure for the detection of herpesvirus reactivations. Moreover, studies should perhaps investigate associations between the amount of reactivated herpesvirus load, the magnitude of IFN system suppression by anti-IFN autoantibodies, immunomodulation induced by clinicians, and multiple relevant patient outcomes (e.g., length of stay in ICU, length of stay in hospital, duration of mechanical ventilation, duration of ARDS, and mortality). We also acknowledge that our study is limited by the inability to assess levels of anti-IFN autoantibodies in patients prior to SARS-CoV-2 infection. Thus, we can currently only speculate that an immunodeficient state was preexisting in certain patients and exacerbated COVID-19 severity and the likelihood of herpesvirus reactivations.

In conclusion, detection of anti-IFN autoantibodies that bind and neutralize the antiviral type I IFNs can be performed relatively easily and rapidly, and could be used in future diagnostic efforts to understand the underlying causes of severe disease in both COVID-19 and other infectious disease manifestations [38]. While there are currently no specific therapies available to counteract the potentially pathogenic activities of anti-IFN autoantibodies, their early diagnosis could be used to stratify "at-risk" individuals for prophylactic vaccinations, or particular drug regimens following infections with certain pathogens, although further evidence would be required to assess benefits of such a strategy. Furthermore, as described here, rapid detection of anti-IFN autoantibodies in ICUs may have diagnostic value in assessing predisposition to potentially detrimental herpesvirus reactivations and thus in prescribing prophylactic therapeutic options to limit their contributions to severe disease.

## Supporting information

**S1 Fig. A multiplexed bead-based assay to detect IFN-binding antibodies. (A)** Schematic representation of the assay principle. Magnetic beads are covalently coated with the indicated

IFNs or albumin as a negative control. Samples are then incubated with the coated beads for 1 h at room temperature to allow binding of any anti-IFN antibodies present. Following wash steps, PE-labeled secondary antibodies specific for antibody isotypes of interest (IgG, IgA, or IgM) are incubated with the beads. After washing, MFI values of bound PE secondary antibodies are measured for each "bead region" on a FlexMap 3D instrument. Schematic created with BioRender.com. **(B, C, and D)** Assay assessment using mouse monoclonal antibodies. IFNα2, IFNβ, IFNω, and albumin-coated beads mixed 1:1:1:1 were incubated with serial dilutions of mouse monoclonal antibodies raised against IFNα2 (B), IFNβ (C), or IFNω (D). Following the assay procedure described in (A), MFI values from IFN-coated beads were obtained and calculated relative to MFI values from albumin-coated beads. Data are representative of at least 2 independent experiments. **(E)** Assay assessment using human plasma samples. IFNα2, IFNβ, IFNω, and albumin-coated beads mixed 1:1:1:1 were incubated with serial dilutions of a pool of healthy donor plasmas (left panel) or a human plasma known to have anti-IFNα2 antibodies (right panel). Following the assay procedure described in (A), MFI values from IFN-coated beads were obtained and calculated relative to MFI values from albumin-coated beads. Data are representative of at least 2 independent experiments. Data underlying this figure can be found in S1 Data. FC, fold change; IFN, interferon; MFI, median fluorescence intensity; PE, phycoerythrin.
(TIF)

**S2 Fig. Analysis of autoantibodies targeting IFNβ in the plasmas and tracheobronchial secretions of critically ill COVID-19 patients.** Multiplexed bead-based assay to detect IgG, IgA, and IgM autoantibodies (autoAbs) against IFNβ in the plasmas (A) or TBSs of COVID-19 ICU patients described in Fig 1A. Pos (positivity) and Neg (negativity) for anti-IFNα2 IgG in plasma samples from the same patient (results from Fig 1A) were used to stratify patients. MFI FC of signal derived from IFN-coated beads relative to the MFI of signal derived from albumin-coated beads is shown. In all panels, red dots indicate the patients/samples that were positive for anti-IFNα2 IgG autoantibodies in plasma (Fig 1A) and are denoted simply for reference. Data underlying this figure can be found in S1 Data. COVID-19, Coronavirus Disease 2019; FC, fold change; ICU, intensive care unit; IFN, interferon; MFI, median fluorescence intensity; TBS, tracheobronchial secretion.
(TIF)

**S1 Data. Raw values underlying the summary data displayed in the figures and tables.**
(XLSX)

## Acknowledgments

We thank Beat M. Frey and the Zurich Blood Transfusion Service of the Swiss Red Cross, Switzerland for permitting access to healthy donor samples. We also thank Georg Kochs (University of Freiburg, Germany) for kindly providing the pGL3-Mx1P-FFluc plasmid. Schematics were created with BioRender.com.

## Author Contributions

**Conceptualization:** Idoia Busnadiego, Irene A. Abela, Silvio D. Brugger, Benjamin G. Hale.

**Data curation:** Irene A. Abela, Pascal M. Frey.

**Formal analysis:** Idoia Busnadiego, Irene A. Abela, Pascal M. Frey, Benjamin G. Hale.

**Funding acquisition:** Irene A. Abela, Benjamin G. Hale.

**Investigation:** Idoia Busnadiego, Irene A. Abela, Pascal M. Frey, Daniel A. Hofmaenner, Silvio D. Brugger, Benjamin G. Hale.

**Methodology:** Idoia Busnadiego, Irene A. Abela.

**Project administration:** Irene A. Abela, Silvio D. Brugger, Benjamin G. Hale.

**Resources:** Daniel A. Hofmaenner, Thomas C. Scheier, Reto A. Schuepbach, Philipp K. Buehler, Benjamin G. Hale.

**Supervision:** Silvio D. Brugger, Benjamin G. Hale.

**Validation:** Idoia Busnadiego.

**Writing – original draft:** Idoia Busnadiego, Irene A. Abela, Pascal M. Frey, Daniel A. Hofmaenner, Silvio D. Brugger, Benjamin G. Hale.

**Writing – review & editing:** Idoia Busnadiego, Irene A. Abela, Pascal M. Frey, Daniel A. Hofmaenner, Thomas C. Scheier, Reto A. Schuepbach, Philipp K. Buehler, Silvio D. Brugger, Benjamin G. Hale.

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
