## [Editor Report · Decision Letter 0]

6 Apr 2022

Dear Dr Hale, 

Thank you for submitting your manuscript entitled "Herpesvirus Reactivations in Critically-Ill COVID-19 Patients with Autoantibodies Neutralizing Type I Interferons" for consideration as a Research Article by PLOS Biology.

Your manuscript has now been evaluated by the PLOS Biology editorial staff, as well as by an academic editor with relevant expertise, and I am writing to let you know that we would like to send your submission out for external peer review.

IMPORTANT: we would consider your submission as a *Discovery Report*. PLOS Biology ‘Discovery Reports’ describe novel and intriguing initial findings with the potential to lead to a significant new result for the field. Discovery Reports are short articles, typically with 2-4 main figures. While the research may be preliminary, studies should be advanced to the stage where observations or findings have been confirmed by independent methods or experimental approaches and obvious alternative interpretations have been ruled out. Please select the Discovery Report article type when you submit your manuscript. 

Once your full submission is complete, your paper will undergo a series of checks in preparation for peer review. Once your manuscript has passed the checks it will be sent out for review. To provide the metadata for your submission, please Login to Editorial Manager (https://www.editorialmanager.com/pbiology) within two working days, i.e. by Apr 08 2022 11:59PM.

If your manuscript has been previously reviewed at another journal, PLOS Biology is willing to work with those reviews in order to avoid re-starting the process. Submission of the previous reviews is entirely optional and our ability to use them effectively will depend on the willingness of the previous journal to confirm the content of the reports and share the reviewer identities. Please note that we reserve the right to invite additional reviewers if we consider that additional/independent reviewers are needed, although we aim to avoid this as far as possible. In our experience, working with previous reviews does save time. 

If you would like to send previous reviewer reports to us, please email me at dummarino@plos.org to let me know, including the name of the previous journal and the manuscript ID the study was given, as well as attaching a point-by-point response to reviewers that details how you have or plan to address the reviewers' concerns. 

Given the disruptions resulting from the ongoing COVID-19 pandemic, please expect some delays in the editorial process. We apologise in advance for any inconvenience caused and will do our best to minimize impact as far as possible.

Kind regards,

Dario

Dario Ummarino, PhD

Senior Editor

PLOS Biology

dummarino@plos.org

---

## [Decision Letter · Decision Letter 1]

20 May 2022

Dear Dr Hale,

Thank you for your patience while your manuscript "Herpesvirus Reactivations in Critically-Ill COVID-19 Patients with Autoantibodies Neutralizing Type I Interferons" was peer-reviewed at PLOS Biology. It has now been evaluated by the PLOS Biology editors, an Academic Editor with relevant expertise, and by several independent reviewers. 

As you will see in the reviews pasted below, two reviewers found your study interesting and important, and raised minor points around methodology, reporting and presentation/interpretation of the results. However, reviewer 3 raised concerns about the strength of the evidence supporting your main findings. Having discussed the reviews with the Academic Editor, we feel that it would not be necessary to address the comments from reviewer 3 regarding ex-vivo models. We appreciate that these additional experiments would strengthen your clinical findings; however, given that we are considering your paper as a Discovery Report, these additional mechanistic details would be beyond the scope of the current submission and may be the subject of future work (to be reported, for instance, in a follow-up Update Article: https://journals.plos.org/plosbiology/s/what-we-publish#loc-update-article). 

In light of the reviews, we are pleased to offer you the opportunity to address the comments from the reviewers in a revised version that we anticipate should not take you very long. Please also make sure to address the following data and other policy-related requests:

1) Title: We would like to suggest a minor modification: "Critically-ill COVID-19 patients with neutralizing autoantibodies against Type I interferons have increased risk of herpesvirus reactivations."

2) Blurb: Please provide a blurb which (if accepted) will be included in our weekly and monthly Electronic Table of Contents, sent out to readers of PLOS Biology, and may be used to promote your article in social media. The blurb should be about 30-40 words long and is subject to editorial changes. It should, without exaggeration, entice people to read your manuscript. It should not be redundant with the title and should not contain acronyms or abbreviations. For examples, view our author guidelines: https://journals.plos.org/plosbiology/s/revising-your-manuscript#loc-blurb

3) Ethics: Please indicate the form of consent obtained (written/oral) or the reason that consent was

not obtained (e.g. the data were analyzed anonymously).

4) Financials: Please confirm that your statement "The author(s) received no specific funding for this work.” is correct. 

5) Data: You may be aware of the PLOS Data Policy, which requires that all data be made available without restriction: http://journals.plos.org/plosbiology/s/data-availability. For more information, please also see this editorial: http://dx.doi.org/10.1371/journal.pbio.1001797

Note that we do not require all raw data. Rather, we ask for all individual quantitative observations that underlie the data summarized in the figures and results of your paper. For an example see here: http://www.plosbiology.org/article/info%3Adoi%2F10.1371%2Fjournal.pbio.1001908#s5

These data can be made available in one of the following forms:

I) Supplementary files (e.g., excel). Please ensure that all data files are uploaded as 'Supporting Information' and are invariably referred to (in the manuscript, figure legends, and the Description field when uploading your files) using the following format verbatim: S1 Data, S2 Data, etc. Multiple panels of a single or even several figures can be included as multiple sheets in one excel file that is saved using exactly the following convention: S1_Data.xlsx (using an underscore).

II) Deposition in a publicly available repository. Please also provide the accession code or a reviewer link so that we may view your data before publication.

Regardless of the method selected, please ensure that you provide the individual numerical values that underlie the summary data displayed in the following figure panels: Figs 1 A-D, 2 ABD, 3, S1B-E, S2 AB, Tables 1, 2, 3.

NOTE: the numerical data provided should include all replicates AND the way in which the plotted mean and errors were derived (it should not present only the mean/average values). Please also review our policies for sharing human research participant data: https://journals.plos.org/plosbiology/s/data-availability#loc-acceptable-data-access-restrictions. We encourage authors to share de-identified or anonymized data. However, when data cannot be publicly shared, we allow authors to make their data sets available upon request. If there are ethical or legal restrictions on sharing a sensitive data set, authors should provide the following information within their Data Availability Statement upon submission: i) Explain the restrictions in detail (e.g., data contain potentially identifying or sensitive patient information); ii) Provide contact information for a data access committee, ethics committee, or other institutional body to which data requests may be sent.

5.1) IMPORTANT: Please also cite the location of the data clearly in each relevant main and supplementary Figure legend, e.g. “Data underlying this Figure can be found in S1 Data”.

5.2) Please ensure that your Data Statement in the submission system accurately describes where your data can be found.

We expect to receive your revised manuscript within three weeks. 

*Published Peer Review History*

*Press*

Sincerely,

Dario

Dario Ummarino, PhD

Senior Editor

PLOS Biology

dummarino@plos.org

Reviewer #1: This paper is excellent. Novel, important, and timely. Suggestions to improve it include:

1/ It is not clear if the entire cohort was tested for neutralization of IFN-alpha and -omega at low and high concentrations (as should be) or only the ELISA-positive patients. It is also not clear either if IFN-beta neutralization was tested.

2/ They should quote all recent studies of auto-Abs to IFN in COVID patients, including Moscow, Japan, Bogota, and Leuven.

3/ They could cite the seminal 1984 paper Ion Gresser, especially because the viral disease was shingles.

4/ Describing the age of AAB+ patients in the COVID cohort in the text would be reader-friendly (it's in the table).

5/ For the HSV/CMV reactivation, a missing info is if the infection had any clinical impact. Was it CMV disease (GI involvement, eye, neuro etc.) or just positive PCR in blood? And if only PCR, which viral load?

6/ About the reactivation of CMV and HSV, it is not entirely clear if those tests were performed only in patients who had positive serologies for those viruses. The term reactivation is used, implying that these patients were infected before.

7/ The authors might tone down a little bit the conclusion, because many parameters were compared between patients with or without AAB. I doubt the association between AAB and viral reactivation (which makes a lot of sense anyway) would remain statistically significant after accounting for multiple testing. The immunological evidence, paradoxically, may be stronger.

Reviewer #2: Busnadiego et al. investigated the interplay between autoantibodies against antiviral type I IFNs, COVID-19, and reactivation of persistent Herpesvirus infections. In about 10% of their over 100 ICU patients with COVID they detected antibodies neutralizing IFN-a2 and/or IFN-o, mostly of the IgG type. The antibodies were present in the plasma as well as in tracheobronchial secretions, indicating they could positively influence SARS-CoV-2 replication in situ. No particular pattern of antibody increase or decrease could be discerned over the observation period. While the association of such anti-IFN antibodies with COVID-19 has already been described, the authors went on to demonstrate that the anti-IFN autoantibodies were also a predictor of Herpes virus (HSV, CMV) reactivation in the COVID patients. 

This important and interesting study shows that the association of anti-IFN autoimmunity with severe COVID-19 may not only (or not necessarily) be based on an elevated SARS-CoV-2 replication, but perhaps because IFN neutralization can reactivate latent Herpes virus infections, a known other COVID risk factor. Moreover, the authors have established a convenient and sensitive neutralizing assay for type I IFNs that will facilitate similar screening efforts in the future. 

The results are therefore advancing our knowledge, will inspire further studies, and may help to improve diagnosis and treatment of COVID-19. I have only minor remarks.

Minor points:

- Apart from the mention that the bead-based assay for autoantibody screening was based on methods from a previous study, the rationale for choosing these three specific IFN subtypes remains unclear.

- Patients were also tested for anti-IFN-b autoantibodies (line 202), but the result of the screen is neither shown nor mentioned. Only the lack of a correlation with IFN-a2 antibodies is described later on (line 236). Similar plots as the ones in Fig.1A and B would be helpful to understand the role (or non-role) anti-IFN-b autoimmunity.

- Why are the detection thresholds for IFN-a2 and IFN-o antibodies set differently? Please provide an explanation.

- Is it known whether the anti-IFN-a2 antisera would also be neutralizing other IFN-a subtypes? Please add at least a short discussion.

- Description of figure 1 in the main text: Please do always provide the number of samples and not only of patients in the text, as it was done in line 219 for the IFN-a2 IgG positives. Otherwise it looks like a discrepancy between the numbers in text and the dots in the figure.

Reviewer #3: Busnadiegoa et al assessed the presence of anti IFN-I ab in severe covid.

They confirmed previous reports and highlight the high prevalence of these AAB in critical COVID-19 partients (around 10%).

They found an association between herpes reactivation and the presence of AAB.

i regret that only a part of patients have been investigated for these analysis.

Moreover, I am not convinced by the direct link between latent virus reactivation and presence of AAB anti IFN-I. 

Ex vivo models are requested to deeper investigate the relationship between herpes virus and IFN impairment.

this would greatly improve the quality of the manuscript.

---

## [Editor Report · Decision Letter 2]

14 Jun 2022

Dear Dr. %LMEAST_NA%,

Thank you for the submission of your revised Discovery Report "Critically-Ill COVID-19 Patients with Neutralizing Autoantibodies against Type I Interferons have Increased Risk of Herpesvirus Disease" for publication in PLOS Biology. On behalf of my colleagues and the Academic Editor, Bill Sugden, I am pleased to say that we can in principle accept your manuscript for publication, provided you address any remaining formatting and reporting issues. These will be detailed in an email you should receive within 2-3 business days from our colleagues in the journal operations team; no action is required from you until then. Please note that we will not be able to formally accept your manuscript and schedule it for publication until you have completed any requested changes.

PRESS

Sincerely, 

Paula

---

Senior Editor

PLOS Biology
